# Visual edge feature enhancement of product appearance design images based on improved retinex algorithm

Cheng-jie Chen[1]*, Guo-rui Tang[2]

1 College of Design & Innovation, Guilin University of Electronic Technology, Beihai Campus, Beihai, China, 2 Dean's Office, Beihai Secondary Vocational and Technical School, Beihai, China

* 18169768107@163.com

## Abstract

Under the influence of complex factors such as lighting, color distortion, and suspended solids, there is a problem of losing edge feature information and blurring edges in product appearance design images. In order to improve the clarity and visual effect of product appearance design, a visual edge feature enhancement method for product appearance design images based on an improved Retinex algorithm is proposed. By using a color correction method based on depth of field estimation, the blue tone of the product appearance design image is removed, and color correction and contrast are applied to the product appearance design image. Improve the Gray Wold algorithm and design an edge attenuation compensation method to solve the problem of edge color attenuation under noise interference, and obtain clearer product appearance design images. On the basis of clarity processing, convert the original RGB image into HSV. On the basis of the Retinex model, multi-level decomposition of brightness is carried out, and different filtering parameters are set to obtain multiple illumination and reflection images with different scale information; Using exponential function and Sigmoid function to process reflection images and illumination images separately, reducing external interference on images of different scales, and solving the difficulty of enhancing images with uneven illumination, high noise, low illumination, and loss of details. At the same time, adaptive nonlinear correction is applied to the saturation component, and the corrected saturation, brightness, and hue are fused and converted into RGB, expanding the edge grayscale feature information in various spatial domains. Improve the weights of traditional bilateral filtering methods, reduce the depth difference between information at different scales, and enhance the visual edge features of product appearance design images. The experimental results show that the proposed method enhances the image with a PCQI of 1.033, an IQE of 0.610, an IQM of 1.830, and an information entropy higher than 0.7. The above data proves that this method has a high richness of edge feature

**Data availability statement:** All relevant data are within the manuscript.

**Funding:** The author(s) received no specific funding for this work.

**Competing interests:** The authors have declared that no competing interests exist.

information after image enhancement, significantly improving the visual edge feature enhancement effect of product appearance design images.

## 1. Introduction

In today's visual dominated era, product appearance design is not only a manifestation of functionality, but also an intuitive display of brand philosophy and aesthetic pursuit. With the rapid development of technology and the increasing aesthetic demands of consumers, how to make product appearance design stand out among many competitors has become the focus of attention for enterprises. The visual effect of product appearance design images, as an important medium for conveying design concepts and product characteristics, directly affects consumers' first impression and purchasing decisions.

Among numerous visual elements, edge features play an indispensable role in the expressive power of product appearance design as the basis for constructing object contours and forms. The clarity, continuity, and uniqueness of the edges directly affect the refinement and recognition of the product's appearance. Therefore, enhancing the visual edge features of product appearance design images has become an important means to improve design quality and enhance market competitiveness.

By using advanced image processing techniques to finely enhance the edge features in the image [1,2], the details of product design can be made more vivid and the form more three-dimensional, thereby generating stronger visual impact and attraction. The above processing methods not only help designers better express their design concepts, but also facilitate consumers to more intuitively understand product features and enhance user experience.

Related researchers have conducted extensive research on the enhancement of visual edge features in product appearance design images [3,4], and have achieved significant research results. Ren and Liu [5] found that there is information redundancy and feature correlation in the spatial neighborhood of polarimetric SAR images, and fully utilizing spatial neighborhood information can help improve the discriminability and robustness of sample features; By introducing an adaptive superpixel clustering algorithm based on polarization statistics HSV color features, a method is proposed to enhance image features using neighborhood correlation. The noise and interference factors present in polarimetric SAR images can cause attenuation of edge color features, leading to instability of color features and ultimately affecting the effectiveness of feature enhancement. Liu et al. [6] proposed using a recurrent generative adversarial network to transform the underwater image enhancement problem into a style transfer problem, achieving unsupervised learning; On the other hand, combining feature decoupling methods to extract style and structural features of the image separately ensures the consistency of the structure before and after enhancement, completing feature enhancement. When processing underwater images, the model is more prone to getting stuck in local optima due to complex factors such as lighting, color distortion,

and suspended solids, resulting in poor image quality. Pang et al. [7] designed a structural feature mapping network and a dual scale feature extraction network. The structural feature mapping network is used to establish global structural feature weights and maintain the spatial structural information of the original image. The dual scale feature extraction network uses multi-scale convolutional layers and fused multi hole convolutions to enhance the network's attention to contextual information, improve its feature extraction ability for regions of interest, and learn feature information at different scales to exchange information between the two scales, generate target enhancement maps, and achieve adaptive enhancement of target region details and textures. When this method is applied to the processing of images with uneven lighting or loss of details, it is easily interfered by redundant reflection images, and the edge enhancement effect of the image is insufficient. Zhao et al. [8] proposed a network shared edge low resolution feature extraction network that effectively extracts features from two edge decoded images with the same content and different details. They also designed a residual recursive compensation network structure and applied it to edge and middle low resolution feature extraction networks. Secondly, a multi description edge upsampling reconstruction network is designed, which adopts a partial network layer parameter sharing strategy. This strategy can reduce the number of network model parameters. A multi description middle path upsampling reconstruction network is proposed, which combines the low resolution features of two edges with the low resolution features of the middle path to achieve deep feature fusion for enhancing multi description compressed images. This method only combines the low resolution features of two edges and the low resolution characteristics of the middle path, and cannot optimize the grayscale features of multiple spatial domains, making it difficult to effectively preserve the edge details of the image. The enhancement effect needs to be improved. In addition to the processing of product design images, the processed images can be used to predict the photovoltaic system illumination and ultraviolet irradiance, detect solar cell defects, and detect the dust condition on the surface of photovoltaic panels. For example, Abdelsatar, M et al. [9] compared the effectiveness of machine learning models such as support vector classification, linear regression, extreme gradient enhancement, gradient enhancement, random forest, and CatBoost, and evaluated their predictive performance in estimating the amount of light and ultraviolet radiation in photovoltaic systems. This method directly utilizes machine learning models for image enhancement, lacking the process of image clarity, and the application effect needs to be improved. Abdelsatar M et al. [10] utilized extensive deep learning techniques for automatic defect recognition in solar cell images. Using 24 different convolutional neural network architectures to classify solar cells into defect and non defect categories. This method did not effectively add to the image before defect category recognition. Abdelsatar M et al. [11] studied the effectiveness of three MobileNet variants based on training image data in correctly classifying dusty and perfect photovoltaic surfaces. This method is limited to the application effect of MobileNet variants, and the enhancement effect is limited.

On the basis of previous research methods, a visual edge feature enhancement method for product appearance design images based on an improved Retinex algorithm is proposed. Improve the Gray Wold algorithm and design an edge attenuation compensation method to solve the problem of edge color attenuation under noise interference, and obtain clearer product appearance design images. Introducing the Retinex algorithm, the image is decomposed into two parts, and fusion and decomposition are carried out separately. Multi scale enhanced reflection images and illumination images are used to reduce different types of interference in images of different scales, and to solve the enhancement problems of images with uneven illumination, high noise, low illumination, and loss of details. Simultaneously optimize the grayscale features of each spatial domain to achieve edge feature enhancement. On the basis of traditional bilateral filtering, enhance the ability to preserve image edge detail information.

## 2. Method for enhancing visual edge features of product appearance design images

### 2.1. Product appearance design, image clarity scheme design

Different product materials have different optical properties, such as the high reflectivity of metals, the texture absorption and scattering of light in wood, and the transparency of plastics, which result in uneven lighting and exhibit different behaviors on different materials. For example, the enamel on the surface of ceramic products

may produce specular reflection, while textiles are more diffuse reflection. This requires the consideration of complex factors such as lighting, color distortion, and suspended solids in the application process of complex images. In general, due to the small distance between the object and the camera, the influence of forward scattering components can be ignored. Therefore, only color correction and contrast enhancement need to be integrated to propose a new solution for clarifying product appearance design images. The image sharpening process is shown in Fig 1.

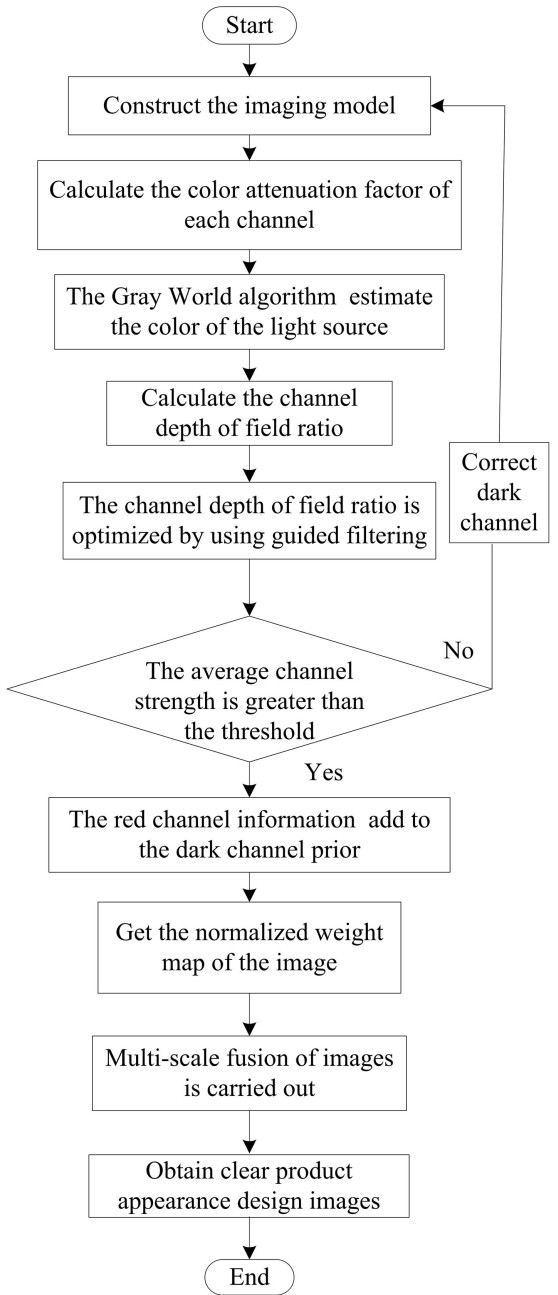

**Fig 1. Image Clarity Processing Flow.**

Simplify the imaging model into the form of formula (1):

$$I(x) = J(x)\,t(x) + B(1 - t(x)) \tag{1}$$

In the above equation, $J(x)\,t(x)$ represents the direct component; $B(1 - t(x))$ represents the background scattering component; $I(x)$ represents the original image; $J(x)$ represents clear images; $B$ represents background light; $t(x)$ represents transmittance, which is expressed in the form of formula (2):

$$t(x) = \exp(-cd(x)) \tag{2}$$

In the above equation, $c$ represents the attenuation coefficient of the medium; $d(x)$ represents the distance between scene point $x$ and the camera.

Due to color cast in product appearance design images, if the background has complex textures or is similar in color to the product, the edges of the product appearance design image will attenuate. To address this, an improved Gray Wold algorithm is introduced to design a compensation method. The main ideas include:

The attenuation process of product appearance design images is represented in the form of formula (2):

$$I_\lambda(x) = J(x)\exp(-z_\lambda d(x)) \tag{3}$$

In the above equation, $\exp(-z_\lambda d(x))$ represents the attenuation factor of light with a wavelength of $\lambda$.

Thus, color correction of product appearance design images can be achieved by estimating and removing the color of the light source.

Calculate the color attenuation factors of each channel and compensate for the color attenuation caused by the medium: that is, treat the blue tone in the image as the color of the light source, use the Gray World algorithm to estimate the color of the light source and remove it. The estimation of light source color can be expressed in the form of formula (4):

$$\left( \frac{\int \left( \frac{I_\lambda(x)}{att_\lambda} \right) dx}{\int I_\lambda(x)\,dx} \right) = a_\lambda e \tag{4}$$

In the above equation, $att_\lambda$ represents the attenuation factor; $I_\lambda(x)$ represents the attenuated image; $a_\lambda$ represents the output gain of each color channel; $e$ represents the color of the light source.

In equation (4), the key to obtaining the attenuation factors of each channel lies in the depth of field of the scene and the estimation of the attenuation coefficient. Therefore, based on the characteristics of the product appearance design image, a depth of field function is proposed to estimate the attenuation differences of each channel, and the attenuation factors of the three channels are calculated separately.

For different scenarios, due to the severe attenuation of red light, the intensity of the red channel in the background is very low, and the scattering of light will result in relatively high intensity of the blue or green channel. Considering the difference in attenuation among the three channels, based on the maximum prior value of the red, blue, and green dark channels, calculate the background light as shown in formula (5):

$$C = \arg\max_x \left( I^{dark(R)}(x) - \max_x \left( I^{dark(G)}(x),\, I^{dark(B)}(x) \right) \right) \tag{5}$$

In the above equation, $I^{dark(R)}(x)$, $I^{dark(G)}(x)$, and $I^{dark(B)}(x)$ represent the dark channel values of the R, G, and B channels of image $I$, respectively.

In order to make the depth of field estimation more accurate, the ratio $\beta_\lambda(x)$ of the global depth of field function $R(x)$ to the depth of field function$_s$ $R_\lambda(x)$ of the R, G, and B component maps is first obtained, which is called the channel depth ratio, that is:

$$\beta_\lambda(x) = \frac{CR(x)}{R_\lambda(x)} \tag{6}$$

Due to the fact that $\beta_\lambda(x)$ is not a constant in practical applications, and its values are also different in R, G, and B channels, the larger the variance, the more relative information it contains. In order to make the depth of field estimation more accurate, more information needs to be obtained from the input image. Therefore, the channel with the largest $\beta_\lambda(x)$ -variance is selected as $\lambda'$, and the depth of field $d_{\lambda'}(x)$ is obtained by combining formula (6):

$$d_{\lambda'}(x) \approx z_{\lambda'}d(x) \approx R_{\lambda'}(x) \approx \frac{R(x)}{\beta_{\lambda'}(x)} \tag{7}$$

In order to restore more details, guided filtering is used to optimize the obtained depth of field function $d_{\lambda'}(x)$ and obtain the attenuation factor of the $\lambda'$ channel. Taking blue light as an example, there are:

$$\begin{cases} d_B(x) = d_{\lambda'}(x) \\ att_B = \exp(-d_B(x)) \end{cases} \tag{8}$$

Based on relevant prior knowledge, the attenuation ratio of red and green light relative to blue light can be calculated, as shown in formula (9):

$$\begin{cases} \dfrac{z_R}{z_B} = \dfrac{b_R A_B}{b_R A_B} \\ \dfrac{z_G}{z_B} = \dfrac{b_G A_B}{b_B A_G} \end{cases} \tag{9}$$

In the above formula, $z_R$, $z_G$, and $z_B$ respectively represent the attenuation factors corresponding to each color channel; $b_R$. $b_G$ and $b_B$ represent the color components corresponding to different color channels, respectively; $A_R$. $A_G$ and $A_B$ represent the attenuation ratio of different color channels.

The attenuation factors $att_R$ and $att_G$ of the other two channels can be obtained from formula (10):

$$\begin{cases} att_R = att_B^{\frac{z_R}{z_B}} \\ att_G = att_B^{\frac{z_G}{z_B}} \end{cases} \tag{10}$$

Substitute formula (10) into formula (4), perform color compensation on the product appearance design image [12], estimate the light source color $e$, and delete it to obtain the color corrected product appearance design image.

In most product appearance design images, certain pixels always have at least one color channel with a very low intensity value, even approaching 0, that is:

$$I^{dark}(x) = \min_{y \in \Omega(x)} \left( \min_{\lambda \in \{R,G,B\}} I_\lambda(x) \right) \to 0 \tag{11}$$

In the above equation, $\Omega(x)$ represents a local block centered around $x$; $I^{dark}(x)$ represents the dark channel value of image $I$. Due to the neglect of the influence of the red channel when estimating the depth of field using dark channel

priors, the dark channel value of the product appearance design image is relatively small, which affects the depth of field estimation. Therefore, it is proposed to set a reasonable threshold for the red channel to determine whether to add red channel information to the dark channel calculation.

Firstly, for the intensity of the red channel, a threshold is set, and then the mean of the red channel is calculated. If the mean is greater than the set threshold, the red channel information is added to the dark channel prior. Otherwise, only the blue-green channel is considered, and the dark channel is corrected to:

$$I^{dark}(x)' = \min\left(\min_{y\in\Omega(x)} pI^R(y)\right), \left(\min_{y\in\Omega(x)} pI^G(y)\right), \left(\min_{y\in\Omega(x)} pI^B(y)\right) \to 0 \tag{12}$$

In the above equation, $y \in \Omega(x)$ represents the image block centered around $x$; $I^R$. $I^G$ and $I^B$ represent the R, G, and B channels in the observed image, respectively; $p$ represents the threshold; $y$ represents the restoration effect.

During the transmission of light, red light attenuates the most severely, while blue light attenuates relatively less, resulting in product design images often presenting a blue tone. Usually, the output $out_R$ of the red channel can be expressed in the form of formula (13):

$$\begin{cases} k_R = \frac{MNMean}{\Sigma R} \\ Mean = \frac{\Sigma R + \Sigma G + \Sigma B}{3MN} \\ out_R = Rk_R = R\frac{\Sigma R + \Sigma G + \Sigma B}{3\Sigma R} \end{cases} \tag{13}$$

In the above equation, $k_R$ represents the output gain of the red channel; $Mean$ represents the average pixel count of the product's exterior design image. By analyzing formula (13), it can be seen that as the value of $R$ approaches 0, it indicates that $out_R$ tends towards infinity, causing the restored image to have a red channel that is not too long, and the processed product appearance design image to appear light red.

Make the gain of each channel positively correlated with its ratio to the intensity values of the three channels in the image. At the same time, in order to fully utilize the information of the red channel in the image, make it inversely correlated with the proportion of that channel.

Firstly, calculate the intensity value $Sum$ of channel 3 in the image as:

$$Sum = \Sigma R + \Sigma G + \Sigma B \tag{14}$$

Then, calculate the ratios $\phi_R$, $\phi_G$, and $\phi_B$ of channels R, G, and B respectively:

$$\begin{cases} \phi_R = \frac{\Sigma R}{Sum} \\ \phi_G = \frac{\Sigma G}{Sum} \\ \phi_B = \frac{\Sigma B}{Sum} \end{cases} \tag{15}$$

Finally, calculate the gains $k'_R$, $k'_G$, and $k'_B$ for channels R, G, and B respectively:

$$\begin{cases} k'_R = 1 - \phi_R = 1 - \frac{\Sigma R}{Sum} = \frac{\Sigma G + \Sigma B}{Sum} \\ k'_G = \frac{\Sigma G}{Sum} \\ k'_B = \frac{\Sigma B}{Sum} \end{cases} \tag{16}$$

The outputs $out_R$, $out_G$, and $out_B$ of channels R, G, and B can be obtained from formula (16):

$$\begin{cases} out_R = Rk'_R = R\frac{\Sigma G + \Sigma B}{\Sigma R + \Sigma G + \Sigma B} \\ out_G = Rk'_G = R\frac{\Sigma G}{\Sigma R + \Sigma G + \Sigma B} \\ out_B = Rk'_B = R\frac{\Sigma B}{\Sigma R + \Sigma G + \Sigma B} \end{cases}$$

(17)

In order to further improve the quality of product appearance design image restoration, feature information of the two input images is extracted [13, 14], and fusion weight maps are defined, namely brightness map, saliency map, local contrast map, and saturation map. The input image information can be adaptively preserved according to the local features of the image, ensuring that the fused image has high brightness, saliency, local contrast, saturation, etc.

Perform adaptive histogram equalization on the three channels of the product appearance design image to obtain input image $I_2$; Using formula (18) to extract features from input images $I_1$ and $I_2$ [15, 16], obtain brightness $W_L^k$, saliency map $W_N^k$, contrast $W_{LC}$, and saturation $W_S$, respectively:

$$\begin{cases} W_L^k = \sqrt{\left[(R^k - L^k) + (G^k - L^k) + (B^k - L^k)^2\right]\Big/3} \\ W_N^k = \sqrt{\left[(L^k - L_m^k) + (a^k - a_m^k) + (b^k - b_m^k)^2\right]\Big/3} \\ W_{LC}(x,y) = \left\| L^k - L_{bc}^k \right\| \\ W_S = \exp\left(-\frac{(S-1)^2}{2\sigma^2}\right) \end{cases}$$

(18)

In the above formula, $L$ represents brightness; $k$ represents the input image number; $a$ and $b$ represent the brightness of input image $I_1$ in the $a$ and $b$ color channels; $a_m^k$ and $b_m^k$ represent the mean values of the $a$ and $b$ color channels; $L_m^k$ represents the average brightness of the Lab space; $L_{bc}^k$ represents the brightness channel obtained after low-pass filtering; $S$ represents the saturation value of each pixel; $\sigma$ represents the standard deviation.

After completing the above operations, obtain the standardized weight maps $W_k$ and $\bar{W}_k$ of the two input images using formula (19):

$$\begin{cases} W_k = W_L^k + W_N^k + W_{LC} + W_S \\ \bar{W}_k = \frac{W_k}{\sum\limits_{k=1}^{2} W_k} \end{cases}$$

(19)

Based on the standardized weight map obtained, perform multi-scale fusion on two input images [17, 18] to obtain a clear product appearance design image $J(x,y)$:

$$J(x,y) = \sum_k \bar{W}_k I_k(x,y)$$

(20)

Therefore, improving the unevenness of lighting caused by light refraction, removing noise, and preserving the sharpness of product appearance design image edges.

## 2.2. Enhancement algorithm based on improved Retinex algorithm

The edge enhancement of product appearance images aims to reduce the grayscale variation in the spatial domain. Therefore, the Retinex algorithm is introduced to decompose the image into two parts, unfold and process them separately, fuse them, and then decompose them to obtain the reflection image and the illumination image [19,20]. Different

sharpness image enhancement functions are used to unfold and enhance from their respective perspectives, while optimizing each spatial domain to complete the enhancement.

According to the Retinex model [21, 22], the product appearance design image is divided into two parts, corresponding to the following expression:

$$U(x, y) = J(x, y) R(x, y) \otimes L(x, y) \tag{21}$$

In the above formula, $U(x, y)$ represents the original product appearance design image; $R(x, y)$ and $L(x, y)$ represent reflection and illumination images, respectively; $\otimes$ represents convolution operation.

In reality, there is no situation where all light is absorbed or reflected by objects. In order to meet the saturation, brightness, and color tone requirements of product appearance design images, the range of the reflection component is set to $R(x, y) \in [0, 1]$, and the dynamic range of the illumination component is obtained as follows:

$$L(x, y) = \frac{U(x, y)}{R(x, y) \geq U(x, y)} \tag{22}$$

The enhancement model for Retinex images is represented in the form of formula (23):

$$U'(x, y) = I(R(x, y)) \otimes g(L(x, y)) \tag{23}$$

In the above formula, $U'(x, y)$ represents the processed product appearance design image; $I(\cdot)$ and $g(\cdot)$ represent functions used to enhance reflection images and illumination images, respectively. The Retinex algorithm is applied to various situations such as uneven lighting, high noise, low illumination, and loss of details. Different enhancement functions can be selectively used for image processing according to the required requirements. Decompose the original image to obtain the decomposed illumination and reflection images, respectively expand and enhance them to obtain the enhanced illumination and reflection images [1, 23], reduce the redundant interference of illumination and reflection images with different scales of information, and fuse them to obtain the final enhanced image.

On the basis of traditional bilateral filtering, the weight $\omega$ is improved to enhance the ability to preserve image edge details. The improved $\omega$ is represented in the form of formula (24):

$$\omega = \begin{cases} 1 - \sqrt{\frac{(i-m)^2 + (j-n)^2}{r}}, & \sqrt{(i-m)^2 + (j-n)^2} \leq r \\ 0, otherwise \end{cases} \tag{24}$$

In the above equation, $r$ represents the size of the filtering window; $m$ represents the size of the product's appearance design image.

Expand the visual edge feature enhancement of product appearance design images [24, 25], and the specific steps are as follows:

(1) Convert the input raw image from RGB model to HSV model through color model, and then extract the saturation $V$ and brightness $S$ from the model.

(2) Using improved bilateral filtering, multi-level decomposition is performed on brightness $V$ to obtain the first, second, and third layer reflection images $R_1, R_2$, and $R_3$ with different scales of information, as well as the third layer illumination image $L_3$.

(3) The reflection image decomposed by Retinex may have insufficient dynamic range due to noise, and it is necessary to expand the dynamic range of the reflection image components through nonlinear transformation, while sharpening details and improving local contrast to restore more realistic surface characteristics. The exponential function has a

non-uniform stretching effect on the reflection component of the input value, which can maintain the enhanced natural transition. By using an exponential function to expand and enhance the decomposed reflection images at different levels, the processed reflection image $f(R_1, R_2, R_3)$ is obtained:

$$f(R_1, R_2, R_3) = \theta_p \otimes V \tag{25}$$

In the above equation, $\theta_p$ represents the indicator function.

(4) The third layer illumination image decomposed by Retinex contains global illumination components with a large dynamic range, but visually sensitive areas are often concentrated in the medium brightness range. The Sigmoid function is a smooth S-shaped curve that can nonlinearly compress a wide range of input brightness values into the [0,1] interval, while enhancing the contrast of the middle brightness region, flexibly controlling the intensity and range of enhancement, and adapting to different lighting conditions. Therefore, using the Sigmoid function to enhance the third layer illumination image $L_3$, the enhanced illumination image $g(L_3)$ is obtained:

$$g(L_3) = \frac{1}{\left(1 + \alpha\sqrt{1 - L_3}/(L_3 + \rho)\right)} \tag{26}$$

In the above equation, $\alpha$ represents the illuminance adjustment coefficient; $\rho$ represents the slack variable.

(5) Integrating the enhanced reflection and illumination images to obtain the enhanced brightness component $V'$:

$$V' = f(R_1, R_2, R_3) \otimes g(L_3) \tag{27}$$

(6) After adaptive nonlinear correction, the processed saturation $S$ is obtained:

$$S = R_n \otimes L_n \tag{28}$$

(7) Integrating $H$, $S$, and $V$ components $H'$, $S'$, and $V'$ to obtain an enhanced image, converting the image back to RGB to obtain the final visual edge feature enhancement result $Z'$:

$$Z' = \frac{I(R_1) \otimes I(R_2) \otimes \cdots I(R_n) \otimes g(L_n)}{H' + S' + V'} \tag{29}$$

The core algorithm steps of the visual edge feature enhancement method for the appearance design image of the above product are pseudocode, as shown in Fig 2.

```
Input: Blurry product image I_RGB with color cast
I_HSV = RGB_to_HSV(I_RGB)   # Color space conversion
βr, βg, βb = estimate_attenuation(I_RGB)   # Depth-based correction
I_corrected = GrayWorld_with_red_threshold(I_RGB, βr, βg, βb)
V, S, H = extract_components(I_HSV)   # Value, Saturation, Hue
R1,R2,R3 = multi_scale_decomp(V)   # Improved bilateral filtering
L3 = V - R3   # Illumination layer
R_enhanced = [exp(R_i) for R_i in [R1,R2,R3]]   # Reflectance enhancement
L_enhanced = sigmoid(L3)   # Illumination enhancement
V' = fuse(R_enhanced, L_enhanced)   # Multi-scale fusion
S' = adaptive_nonlinear(S)   # Saturation correction
Output = HSV_to_RGB(V', S', H)   # Final enhanced image
```

**Fig 2. Algorithm pseudocode.**

## 3. Experiment

Using a product appearance design image acquisition device, sample 500 product appearance design images for expansion search and training, and select 100 images for expansion testing, as shown in Fig 3.

The experimental data of 600 images in the dataset is sourced from the industrial grade image acquisition device for product appearance design shown in Fig 3, which uses a 25 megapixel CMOS sensor. The experimental data of 600 images includes products of bottled, bagged, barreled, boxed and other types, with materials including paper, plastic, etc. The data has differentiated and diverse characteristics. Each product includes front view, oblique view, side view and other angles, covering the complete appearance features of the product's design image. The original image is infused with interference factors such as reflective spots, occlusion, and texture blur, which can demonstrate the enhanced robustness of our method. Therefore, it is necessary for application.

In the Matlab platform, the software and hardware development environment is as follows:

(1) The experimental hardware platform is Intel (R) Core (TM) i7-8700KCPU 3 70 UHz processor, 16 A desktop computer with 00 UB RAM and 64 bit operating system.

(2) The image size is 256×256×3, with a resolution of 25 million pixels, a frame rate of 20FPS, an exposure time of 1/60s, and a focal length of 30mm.

(3) Based on the number of images and resolution characteristics, in order to fully train the algorithm, the number of iterations is set to 200.

(4) Improve Gray World algorithm parameters: use Gaussian attenuation function, standard deviation σ=1.5, compensation radius r=3.

Retinex multi-scale decomposition level: Level 3. Gaussian filtering parameters at all levels: Layer 1 (coarse scale): Window size 31×31 (capturing global illumination). Layer 2 (Medium Scale): Window size 15×15 (Medium Structure). Layer 3 (fine scale): Window size 7×7 (retaining high-frequency details).

(5) Running time (single 1080P image): CPU mode: approximately 0.8 seconds. GPU acceleration: about 0.3 seconds (achieving bilateral filtering and Retinex decomposition). Multi scale decomposition adopts multi threading, and each level of filtering is independently calculated.

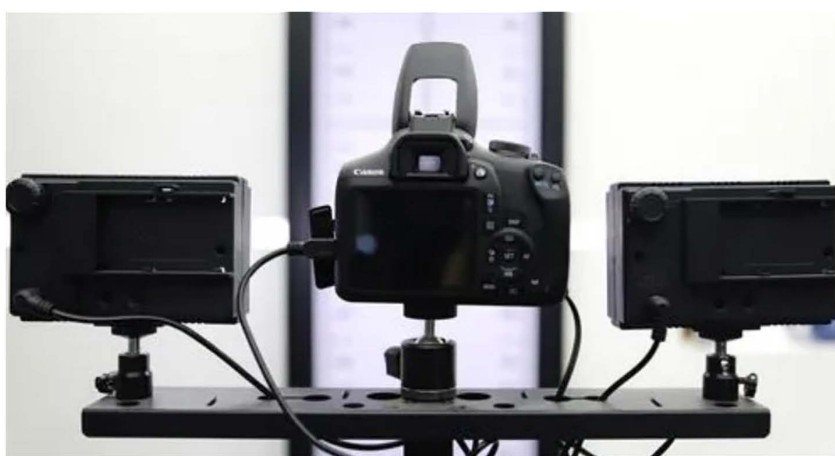

**Fig 3. Product appearance design image collection environment.**

Based on existing experience, bilateral filtering weights $\in \{3, 5, 7, 9, 11, 13, 15\}$ were set and modified for verification. By calculating the feature retention rate under different filtering weights, it was found that when the weight was greater than 5, the feature retention rate changed, effectively suppressing the detail blurring phenomenon caused by large window filtering. Considering the computational complexity, we have chosen a bilateral filtering weight of 7.

The experimental sample is shown in Fig 4.

Experiment 1: Clarity Performance Test.

In order to verify the superiority of the proposed method, a comparison was made between the physical model based method and the Tetrolet transform based method. Four different product appearance design images were selected for testing, and the experimental results are shown in Fig 5.

From Fig 5, it can be seen that the background of the original image sample has complex textures and similar texture colors. Therefore, during the image transmission process, the red light attenuation is more severe, and the product appearance design image presents a blue tone color deviation, with not only more noise but also blurred edges. The method proposed in this article corrects the red and blue channel values using a depth estimation function, and designs an edge attenuation compensation method using the grey world algorithm. By estimating the color of the light

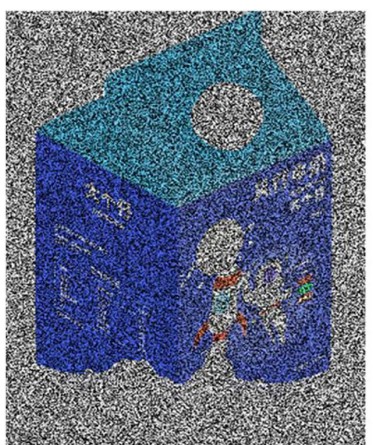
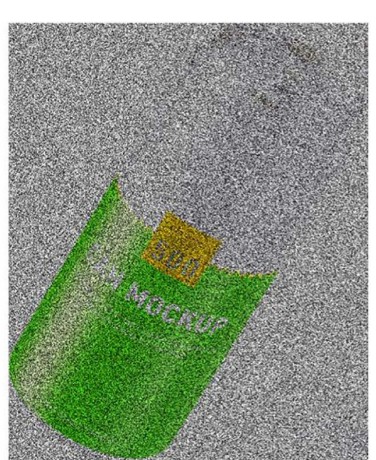
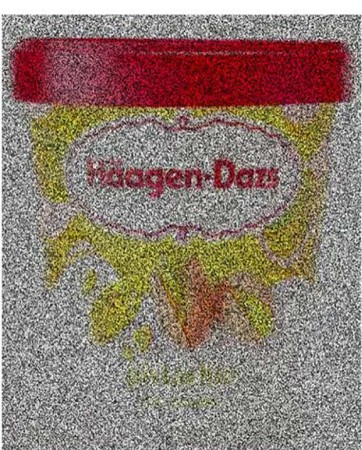
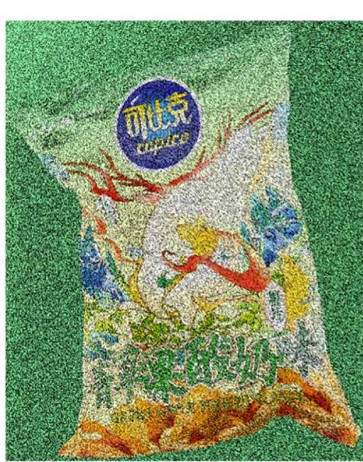

**Fig 4. Sample of product appearance design images.**

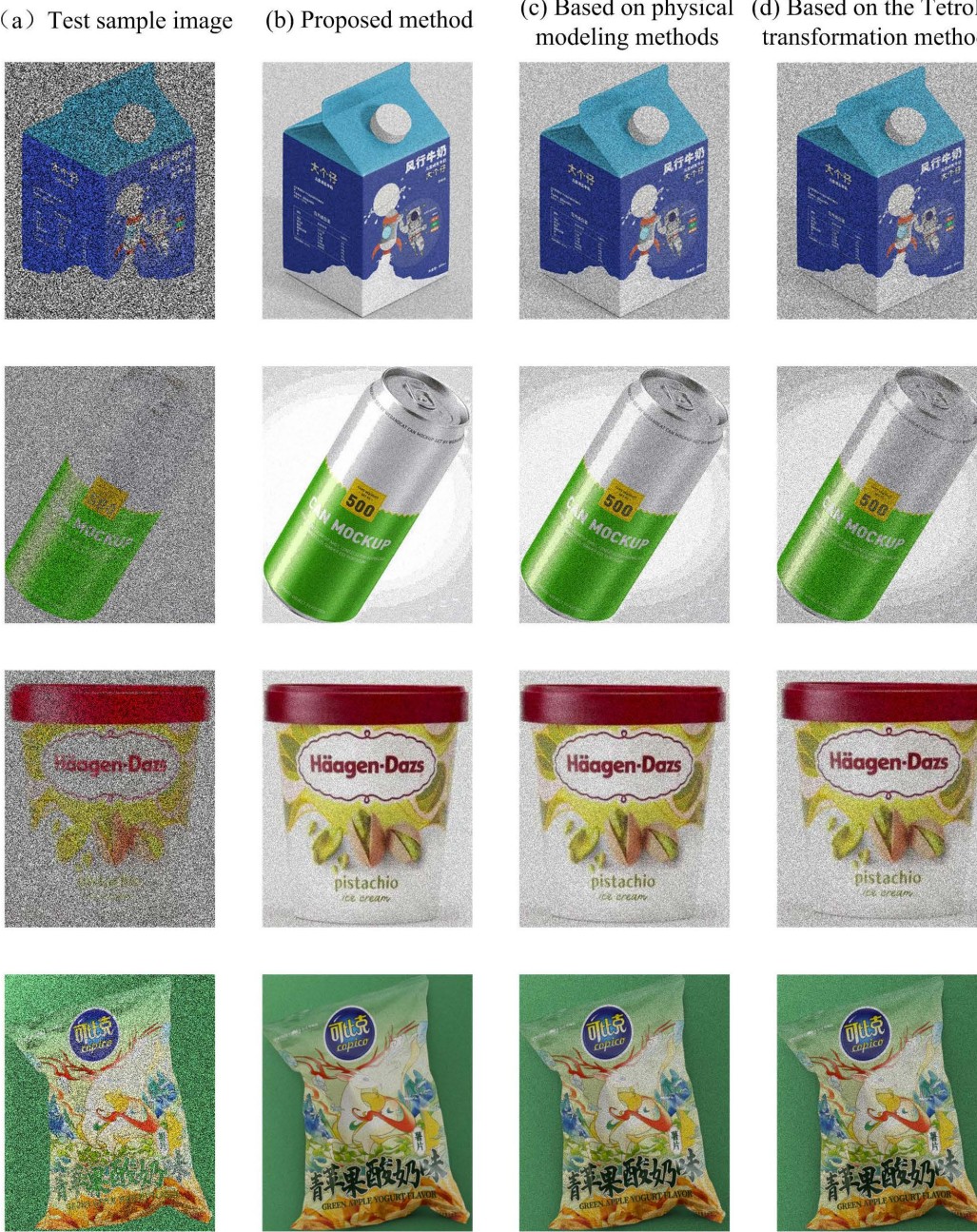

（a）Test sample image　(b) Proposed method　(c) Based on physical modeling methods　(d) Based on the Tetrolet transformation method

**Fig 5. Comparison of image clarity results of product appearance design using different methods.**

source, the color attenuation of the product appearance design image is compensated and corrected to remove blue color cast and noise, making the texture edges of the image clearer. However, other methods neglect the depth difference of different color channels and do not compensate for color attenuation, resulting in relatively poor image clarity. Therefore, the image clarity results demonstrate the effectiveness of the fusion of depth estimation and grey world algorithm in this paper.

Use four evaluation metrics, namely Block Contrast Quality Index (PCQI), Image Quality Evaluation (IQE), Image Quality Measurement (IQM), and Clarity Time, to evaluate the performance of each method. Among them, PCQI is used to measure the contrast of general degraded images, IQE is used to measure the comprehensive indicators of color cast, blur, and contrast of images, and IQM is used to measure the comprehensive indicators of color, clarity, and contrast of images. The higher the values of PCQI, IQE, and IQM indicators, the higher the quality of the image. The lower the clarity time indicator value, the higher the computational efficiency of the method. Table 1 presents the mean, median, and variance values of the evaluation indicators for the processing results of four images using the aforementioned methods.

From Table 1, it can be seen that the PCQI, IQE, and IQM values of the proposed method are higher than the other two methods, indicating that the clarity performance of the proposed method is the best, which can significantly improve the clarity of product appearance design images, and the details of the images are also more prominent, with good visual effects. The calculation efficiency of this method is higher than that of other methods.

**Table 1. Comparison of PCQI, IQE, and IQM experimental results using different methods.**

| Test image number | Test indicators | test method | | |
|---|---|---|---|---|
| | | Proposed method | Based on physical modeling methods | Based on the Tetrolet transformation method |
| 01 | PCQI | 1.033(Median:1.032, Variance:0.0001) | 0.889(Median:0.888, Variance:0.0002) | 0.782(Median:0.781, Variance:0.0001) |
| | IQE | 0.602(Median:0.601, Variance:0.0001) | 0.595(Median:0.594, Variance:0.0002) | 0.579(Median:0.578, Variance:0.0002) |
| | IQM | 1.640(Median:1.639, Variance:0.0003) | 1.633(Median:1.631, Variance:0.0001) | 1.604(Median:1.602, Variance:0.0001) |
| | Clarity Time/ms | 13.671(Median:13.669, Variance:0.0002) | 19.392(Median:19.395, Variance:0.0002) | 23.341(Median:23.340, Variance:0.0002) |
| 02 | PCQI | 1.007(Median:1.006, Variance:0.0001) | 0.920(Median:0.923, Variance:0.0001) | 0.885(Median:0.883, Variance:0.0001) |
| | IQE | 0.610(Median:0.609, Variance:0.0002) | 0.597(Median:0.593, Variance:0.0002) | 0.591(Median:0.590, Variance:0.0001) |
| | IQM | 1.773(Median:1.772, Variance:0.0001) | 1.760(Median:1.757, Variance:0.0001) | 1.748(Median:1.747, Variance:0.0002) |
| | Clarity Time/ms | 11.892(Median:11.891, Variance:0.0002) | 17.983(Median:17.979, Variance:0.0002) | 20.194(Median:20.192, Variance:0.0002) |
| 03 | PCQI | 1.020(Median:1.021, Variance:0.0001) | 0.923(Median:0.922, Variance:0.0001) | 0.891(Median:0.889, Variance:0.0001) |
| | IQE | 0.598(Median:0.597, Variance:0.0002) | 0.590(Median:0.591, Variance:0.0004) | 0.582(Median:0.581, Variance:0.0004) |
| | IQM | 1.830(Median:1.831, Variance:0.0002) | 1.757(Median:1.756, Variance:0.0001) | 1.719(Median:1.717, Variance:0.0002) |
| | Clarity Time/ms | 12.006(Median:12.008, Variance:0.0001) | 18.692(Median:18.691, Variance:0.0002) | 21.359(Median:21.357, Variance:0.0001) |
| 04 | PCQI | 1.005(Median:1.003, Variance:0.0002) | 0.884(Median:0.882 Variance:0.0001) | 0.803(Median:0.801, Variance:0.0002) |
| | IQE | 0.577(Median:0.576, Variance:0.0001) | 0.523(Median:0.521 Variance:0.0002) | 0.504(Median:0.501, Variance:0.0001) |
| | IQM | 1.652(Median:1.651, Variance:0.0003) | 1.536(Median:1.535 Variance:0.0001) | 1.515(Median:1.511, Variance:0.0002) |
| | Clarity Time/ms | 12.837(Median:12.838, Variance:0.0001) | 19.252(Median:19.250 Variance:0.0002) | 21.015(Median:21.014, Variance:0.0001) |

Experiment 2: Performance testing of enhancing visual edge features in product appearance design images:

In order to further verify the effectiveness of the proposed method, different methods were used in the experiment to enhance the visual edge features of product appearance design images. The experimental results obtained are shown in Fig 6

As shown in Fig 6, after the proposed method is used to enhance the visual edge features of the product appearance design image, the image is more in line with human vision, and the edges of the image are clearer, effectively preserving

(a) Test sample image    (b) Proposed method    (c) Feature decoupling method    (d) Parallel Multi Feature Extraction Network Method

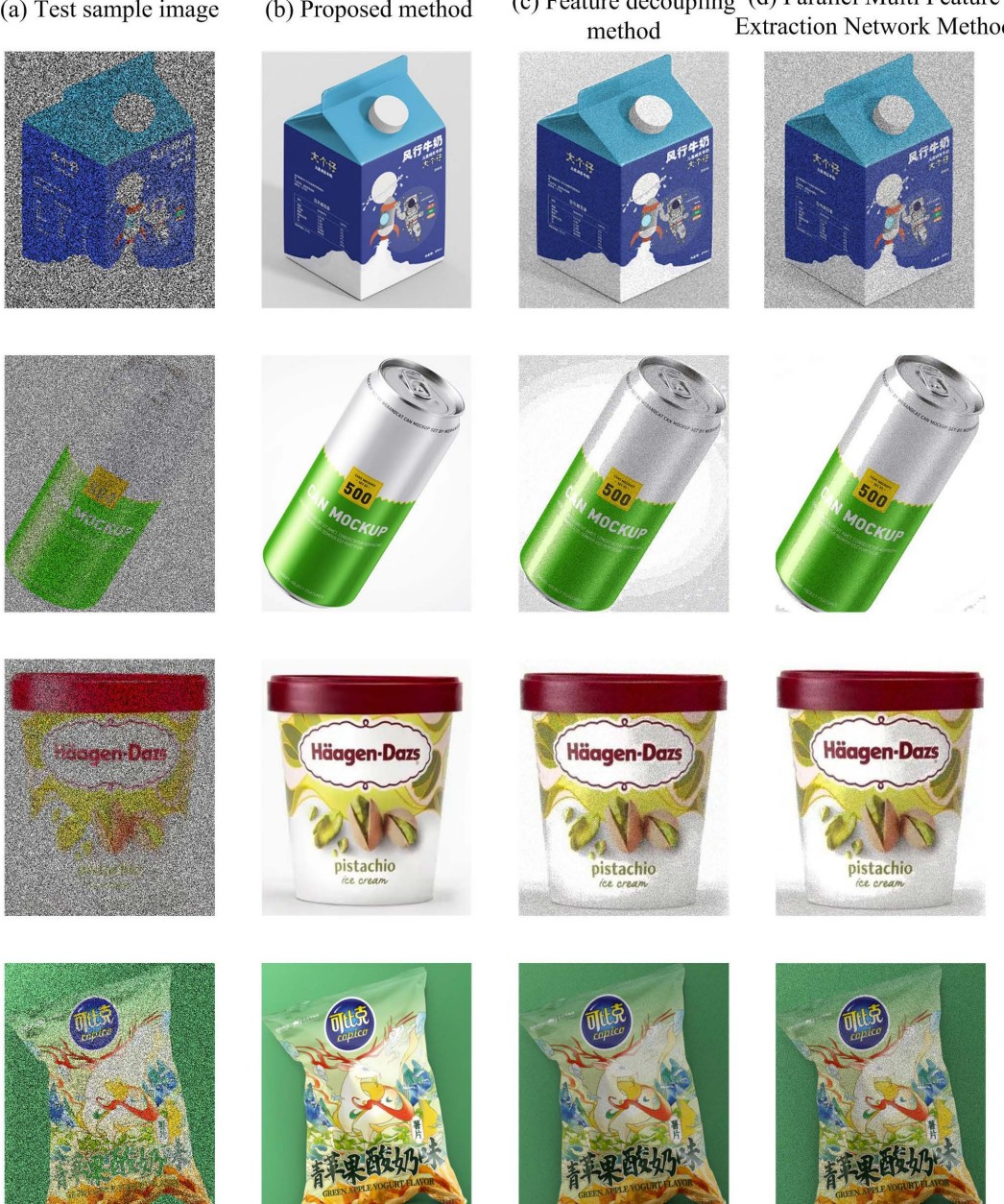

**Fig 6. Comparison of visual edge feature enhancement results of product appearance design images using different methods.**

more detailed information; However, after the other two methods were enhanced, the image still had blurriness and color cast, and the overall enhancement effect was unsatisfactory. It can be seen that the proposed method enhances the edge features of product appearance images more effectively.

The richness of visual edge feature enhancement in product appearance design images is measured using Entropy. The larger the Entropy value, the more information is contained in the image. Add modern deep learning methods currently leading image enhancement, such as GAN-based enhancement methods as a contrast method. The specific experimental test results are shown in Fig 7.

From Fig 7, it can be seen that the information entropy of the proposed method is the highest among the three methods, indicating that the proposed method can achieve more satisfactory image visual edge feature enhancement effects and significantly improve the richness of information in the image.

Design ablation experiments to verify the contributions of key steps in the article, such as depth based color correction, weight map fusion, improved bilateral filters, exponential functions, and S-shaped functions. The information entropy comparison results of different steps are shown in Table 2.

According to Table 2, the absence of each key step leads to the loss of image edge feature information, resulting in a decrease in image information entropy. Therefore, it is necessary to improve the results through depth based color correction, weight map fusion, improved bilateral filters, exponential functions, and S-shaped functions.

## 4. Conclusion

Considering the clarity and visual effect defects of traditional product appearance design images, a method for enhancing visual edge features of product appearance design images based on an improved Retinex algorithm is proposed. Obtain the following conclusion:

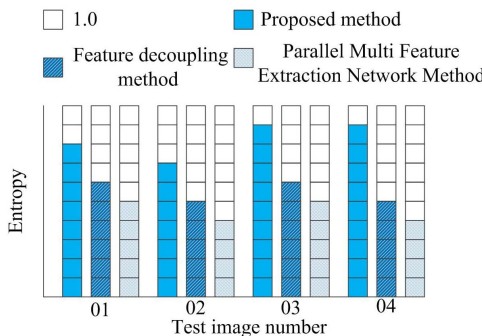

**Fig 7. Comparison of experimental results of information entropy using different methods.**

**Table 2. Ablation Experiment**

| Method steps | Entropy of information |
|---|---|
| Color correction + weight map fusion + improved bilateral filter + exponential function and S-shaped function | 0.9 |
| Color correction + weight map fusion + improved bilateral filter | 0.7 |
| Color correction + weight map fusion | 0.6 |
| Color correction | 0.3 |
| Color correction + improved bilateral filter | 0.5 |

(1) By using a color correction method based on depth estimation, the processed saturation, brightness, and hue are fused and converted into RGB, reducing the redundancy of lighting and reflection images with different scales of information while reducing the depth difference between different scales of information. The clarity is significantly improved, and the block contrast quality index (PCQI), image quality evaluation (IQE), and image quality measurement (IQM) are also significantly improved, resulting in better overall clarity performance.

(2) By using exponential function and Sigmoid function to process the reflection image and illumination image respectively, and adaptively non-linear correction of the saturation component, the overall quality is significantly improved while reducing the loss of edge feature information. This can effectively preserve more detailed information in the image, and the overall enhancement effect is more stable. At the same time, the increase in information entropy is also quite significant.

Although the proposed method has achieved significant research results, there are still a series of shortcomings. The following provides future work prospects for the proposed method:

(1) In the application process of Retinex algorithm, it is of great research value to achieve a balance between filtering effect, refinement of enhancement processing, and computation time, while ensuring that all indicators are relatively optimal. In the future, further research will be conducted in this area.

(2) The visual system of the human eye is currently the best object imaging processing system in the field of image processing, and the Retinex algorithm used in the proposed method only imitates a part of the features of the human eye's visual system that can be practically applied and integrated before being published. How to simulate the human visual system more accurately and accurately, and meet the needs of digital image processing in various scenarios, has become the primary focus of future research.

## Author contributions

**Conceptualization:** Cheng-jie Chen.

**Data curation:** Guo-rui Tang.

**Formal analysis:** Guo-rui Tang.

**Methodology:** Cheng-jie Chen.

**Project administration:** Cheng-jie Chen.

**Software:** Cheng-jie Chen.

**Supervision:** Guo-rui Tang.

**Writing – original draft:** Cheng-jie Chen, Guo-rui Tang.

**Writing – review & editing:** Cheng-jie Chen, Guo-rui Tang.

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
