## [Decision Letter · Decision Letter 0]

15 May 2025

PONE-D-25-02402

Visual Edge Feature Enhancement of Product Appearance Design Images Based on Improved Retinex Algorithm

PLOS ONE

Dear Dr. Chen,

Thank you for submitting your manuscript to PLOS ONE. After careful consideration, we feel that it has merit but does not fully meet PLOS ONE’s publication criteria as it currently stands. Therefore, we invite you to submit a revised version of the manuscript that addresses the points raised during the review process.

We look forward to receiving your revised manuscript.

Kind regards,

Yongjie Li

Academic Editor

PLOS ONE

Journal Requirements:

Additional Editor Comments:

Comments from the Editorial Office: One or more of the reviewers has recommended that you cite specific previously published works. Members of the editorial team have determined that the works referenced are not directly related to the submitted manuscript. As such, please note that it is not necessary or expected to cite the works requested by the reviewer.

Reviewers' comments:

Reviewer's Responses to Questions

**Comments to the Author**

1. Is the manuscript technically sound, and do the data support the conclusions?

Reviewer #1: Yes

Reviewer #2: Yes

2. Has the statistical analysis been performed appropriately and rigorously? 

Reviewer #1: Yes

Reviewer #2: N/A

3. Have the authors made all data underlying the findings in their manuscript fully available?

Reviewer #1: Yes

Reviewer #2: Yes

4. Is the manuscript presented in an intelligible fashion and written in standard English?

Reviewer #1: Yes

Reviewer #2: Yes

5. Review Comments to the Author

Reviewer #1: 1. Abstract: The abstract is overly technical and lacks clarity on the novelty of the method. It should include quantifiable improvements and avoid subjective terms like "better and more beautiful."

2.. Introduction: The research gap is unclear. A critical comparison with existing methods is missing. Additionally, cite these papers to highlight the broader impact of image processing beyond product design:

1- Machine Learning-Based Prediction of Illuminance and Ultraviolet Irradiance in Photovoltaic Systems., 2- Automated Defect Detection in Solar Cell Images Using Deep Learning Algorithms., 3- Detecting Dusty and Clean Photovoltaic Surfaces Using MobileNet Variants for Image Classification.

3. Related Work: The literature review is descriptive rather than analytical. It lacks a quantitative comparison of previous approaches and their limitations.

4. Methodology (Section 2.1):

The depth of field estimation and Gray World algorithm are not well justified.

There is no clear explanation or diagram for the algorithm’s workflow.

Mathematical derivations lack validation or comparative analysis.

5. Methodology (Section 2.2):

The Retinex modification is unclear—how is it improved over standard methods?

The choice of exponential and Sigmoid functions for enhancement is not justified.

The bilateral filtering modification lacks empirical validation.

6. Experimental Setup:

No computational efficiency analysis or comparison with existing benchmarks.

The choice of 200 iterations is arbitrary and should be explained.

Reviewer #2: 1. Baseline choices such as “physical modelling method”, “Tetrolet transform” are outdated. I suggest the authors include state-of-the-art learning-based methods a few of which include RetinexNet, Zero-DCE, SCI, MIRNet-v2, U-Retinex.

2. There is no ablation study verifying the contribution of each component (depth-based color correction, weight-map fusion, improved bilateral filter, exponential vs. sigmoid functions, etc.). Including such results may improve the results.

3. The manuscript lacks comparison with modern deep-learning approaches (e.g., GAN-based or CNN-based enhancement) that currently dominate image enhancement literature. Without such benchmarks, it is unclear if the proposed algorithm advances the state of the art beyond classical methods

4. The origin, diversity, and availability of the 500/100 image dataset are not described. The authors should include such descriptions in the manuscript.

5. Claims such as “significantly improve … bring better and more beautiful product appearance design results” in the Abstract are qualitative and unsupported.

6. Key hyper-parameters such as window size , decomposition levels etc., are not reported; neither are hardware, running time, nor implementation details

7. Please also correct grammatical errors in the paper as there were a few.

6. PLOS authors have the option to publish the peer review history of their article (what does this mean? ). If published, this will include your full peer review and any attached files.

**Do you want your identity to be public for this peer review?** For information about this choice, including consent withdrawal, please see our Privacy Policy .

Reviewer #1: No

Reviewer #2: No

---

## [Author Response · Author response to Decision Letter 1]

9 Jun 2025

Modification instructions:

Reviewer Modification instructions for #1

1. Abstract: The abstract is overly technical and lacks clarity on the novelty of the method. It should include quantifiable improvements and avoid subjective terms like "better and more beautiful."

Re: The abstract has been revised to highlight innovative improvements, such as improving the Gray Wold algorithm, designing an edge attenuation compensation method to solve the problem of edge color attenuation under noise interference, and obtaining clearer product appearance design images. Improve the weights of traditional bilateral filtering methods and reduce the depth of field differences between information at different scales. At the same time, increasing quantitative data reflects the advantages of the proposed method: the enhanced image PCQI reaches 1.033, IQE reaches 0.610, IQM reaches 1.830, and information entropy is higher than 0.7.

2. Introduction: The research gap is unclear. A critical comparison with existing methods is missing. Additionally, cite these papers to highlight the broader impact of image processing beyond product design:

1- Machine Learning-Based Prediction of Illuminance and Ultraviolet Irradiance in Photovoltaic Systems., 2- Automated Defect Detection in Solar Cell Images Using Deep Learning Algorithms., 3- Detecting Dusty and Clean Photovoltaic Surfaces Using MobileNet Variants for Image Classification.

Re: The three research results you listed have been compared with the method proposed in this paper. The existing method can predict the illumination and ultraviolet irradiance of photovoltaic systems, detect defects in solar cells, and detect dust on the surface of photovoltaic panels using processed images. But these three methods focus on the application of images and ignore the enhancement processing effect of images. This paper's method can widen the gap with these methods in image addition methods.

3. Related Work: The literature review is descriptive rather than analytical. It lacks a quantitative comparison of previous approaches and their limitations.

Re: The analytical content of the literature review has been supplemented to analyze the limitations of existing methods: The noise and interference factors present in polarimetric SAR images can cause attenuation of edge color features, leading to instability of color features and ultimately affecting the effectiveness of the feature enhancement method proposed in reference [5]. When processing underwater images, the model proposed in reference [6] is more prone to getting stuck in local optima due to complex factors such as lighting, color distortion, and suspended solids, resulting in poor image quality. When the method described in reference [7] is applied to the processing of images with uneven lighting or loss of details, it is easily interfered by redundant reflection images, and the edge enhancement effect of the image is insufficient. The method in reference [8] only combines the low resolution features of two edges and the low resolution characteristics of the middle path, which cannot optimize the grayscale features of multiple spatial domains and effectively preserve the edge details of the image. The enhancement effect needs to be improved.

In response to the shortcomings of the existing methods mentioned above, this paper proposes a solution. The difference between this paper's method and existing methods lies in the improvement of the Gray Wold algorithm, the design of an edge attenuation compensation method to solve the problem of edge color attenuation under noise interference, and the acquisition of clearer product appearance design images. Introducing the Retinex algorithm, the image is decomposed into two parts, and fusion and decomposition are carried out separately. Multi scale enhanced reflection images and illumination images are used to reduce different types of interference in images of different scales, and to solve the enhancement problems of images with uneven illumination, high noise, low illumination, and loss of details. Simultaneously optimize the grayscale features of each spatial domain to achieve edge feature enhancement. On the basis of traditional bilateral filtering, enhance the ability to preserve image edge detail information.

4. Methodology (Section 2.1):

The depth of field estimation and Gray World algorithm are not well justified.

Re: This paper applied the depth of field estimation function and grey world algorithm to clarify the image, and the experimental results in Figure 3 can prove the effectiveness of the depth of field estimation function and grey world algorithm. As shown in Figure 3, the background of the original image sample has complex textures and similar texture colors. Therefore, during the image transmission process, the red light attenuation is more severe, and the product appearance design image presents a blue tone color deviation, with not only more noise but also blurred edges. The method proposed in this paper corrects the red and blue channel values using a depth estimation function, and designs an edge attenuation compensation method using the grey world algorithm. By estimating the color of the light source, the color attenuation of the product appearance design image is compensated and corrected to remove blue color cast and noise, making the texture edges of the image clearer.

There is no clear explanation or diagram for the algorithm’s workflow.

Re: The workflow diagram of the algorithm has been supplemented clearly. Please see Figure 1 for details.

Mathematical derivations lack validation or comparative analysis.

Re: The advantages of the mathematical derivation process of depth estimation and grey world algorithm compared to other methods have been compared through Figure 3. Figure 3 compares the clarity results of product appearance design images using different methods. This paper uses a depth estimation function to correct the red and blue channel values, and designs an edge attenuation compensation method using the grey world algorithm. By estimating the color of the light source, the color attenuation of the product appearance design image is compensated and corrected to remove blue color cast and noise, making the image texture edges clearer. However, other methods neglect the depth difference of different color channels and do not compensate for color attenuation, resulting in relatively poor image clarity. Therefore, the image clarity results demonstrate the effectiveness of the mathematical derivation that combines depth estimation and grey world algorithm in this paper.

5. Methodology (Section 2.2):

The Retinex modification is unclear—how is it improved over standard methods?

Re: The improvement of Retinex algorithm compared to conventional standard methods is that it can selectively use different enhancement functions for image processing according to the required requirements. Decompose the original image to obtain the decomposed illumination and reflection images, and enhance them separately to obtain the enhanced illumination and reflection images, reducing the redundant interference of illumination and reflection images with different scales of information. After fusing the enhanced illumination and reflection images, the enhancement problems of uneven illumination, high noise, low illumination, and loss of details in the image can be solved.

In addition to the Retinex algorithm, this method improves the filtering weights on the basis of traditional bilateral filtering, obtaining information at different scales and enhancing the ability to preserve image edge details.

The choice of exponential and Sigmoid functions for enhancement is not justified.

Re: The reflection image decomposed by Retinex may have insufficient dynamic range due to noise, and it is necessary to expand the dynamic range of the reflection image components through nonlinear transformation, while sharpening details and improving local contrast to restore more realistic surface characteristics. The exponential function has a non-uniform stretching effect on the reflection component of the input value, which can maintain the enhanced natural transition. Therefore, an exponential function is used for image enhancement. The third layer illumination image decomposed by Retinex contains global illumination components with a large dynamic range, but visually sensitive areas are often concentrated in the medium brightness range. The Sigmoid function is a smooth S-shaped curve that can nonlinearly compress a wide range of input brightness values into the [0,1] interval, while enhancing the contrast of the middle brightness region, flexibly controlling the intensity and range of enhancement, and adapting to different lighting conditions. Therefore, the Sigmoid function is applied to enhance the third layer illumination image after Retinex decomposition.

The bilateral filtering modification lacks empirical validation.

Re: Additional experience verification has been added: Based on existing experience, bilateral filtering weights ∈ {3, 5, 7, 9, 11, 13, 15} were set and modified for verification. By calculating the feature retention rate under different filtering weights, it was found that when the weight was greater than 5, the feature retention rate changed, effectively suppressing the detail blurring phenomenon caused by large window filtering. Considering the computational complexity, we have chosen a bilateral filtering weight of 7.

6. Experimental Setup:

No computational efficiency analysis or comparison with existing benchmarks.

Re: The calculation efficiency of each method has been verified through clear time evaluation indicators. The lower the clarity time indicator value, the higher the computational efficiency of the method. The comparison results are shown in Table 1.

The choice of 200 iterations is arbitrary and should be explained.

Re: The selection of 200 iterations is not arbitrary. The explanation is as follows: Based on the number of images and resolution characteristics, in order to fully train the algorithm, the number of iterations is set to 200.

Reviewer Modification instructions for #2

1. Baseline choices such as “physical modelling method”, “Tetrolet transform” are outdated. I suggest the authors include state-of-the-art learning-based methods a few of which include RetinexNet, Zero-DCE, SCI, MIRNet-v2, U-Retinex.

Re: This paper adopts the Retinex algorithm, which is in line with the most advanced learning methods. In addition, this paper innovatively improves the Gray Wold algorithm and traditional bilateral filtering methods. In order to demonstrate the advanced nature of this method, the results in Figure 5 show that the image enhancement effect of the proposed method is better than that of the classical latest method.

2. There is no ablation study verifying the contribution of each component (depth-based color correction, weight-map fusion, improved bilateral filter, exponential vs. sigmoid functions, etc.). Including such results may improve the results.

Re: Design ablation experiments to verify the contributions of key steps in the paper, such as depth based color correction, weight map fusion, improved bilateral filters, exponential functions, and S-shaped functions. The information entropy comparison results of different steps are shown in Table 2. According to Table 2, the absence of each key step leads to the loss of image edge feature information, resulting in a decrease in image information entropy. Therefore, it is necessary to improve the results through depth based color correction, weight map fusion, improved bilateral filters, exponential functions, and S-shaped functions.

3. The manuscript lacks comparison with modern deep-learning approaches (e.g., GAN-based or CNN-based enhancement) that currently dominate image enhancement literature. Without such benchmarks, it is unclear if the proposed algorithm advances the state of the art beyond classical methods

Re: To demonstrate whether the method proposed in this paper surpasses the latest techniques of classical methods, add modern deep learning methods currently leading image enhancement, such as GAN-based enhancement methods as a contrast method. The specific experimental test results are shown in Figure 5. The results in Figure 5 show that the image enhancement effect of the proposed method is better than that of the classical latest method.

4. The origin, diversity, and availability of the 500/100 image dataset are not described. The authors should include such descriptions in the manuscript.

Re:The experimental data of 600 images in the dataset is sourced from the industrial grade image acquisition device for product appearance design shown in Figure 2, which uses a 25 megapixel CMOS sensor. The experimental data of 600 images includes products of bottled, bagged, barreled, boxed and other types, with materials including paper, plastic, etc. The data has differentiated and diverse characteristics. Each product includes front view, oblique view, side view and other angles, covering the complete appearance features of the product's design image. The original image is infused with interference factors such as reflective spots, occlusion, and texture blur, which can demonstrate the enhanced robustness of our method. Therefore, it is necessary for application.

5. Claims such as “significantly improve … bring better and more beautiful product appearance design results” in the Abstract are qualitative and unsupported.

Re: Specific qualitative descriptions and quantitative data support have been added to the abstract: The experimental results show that the proposed method enhances the image with a PCQI of 1.033, an IQE of 0.610, an IQM of 1.830, and an information entropy higher than 0.7. The above data proves that this method has a high richness of edge feature information after image enhancement, significantly improving the visual edge feature enhancement effect of product appearance design images.

6. Key hyper-parameters such as window size , decomposition levels etc., are not reported; neither are hardware, running time, nor implementation details

Re: Experimental hardware settings: The experimental hardware platform is Intel (R) Core (TM) i7-8700KCPU 3 70 UHz processor, 16 A desktop computer with 00 UB RAM and 64 bit operating system.

Algorithm implementation details settings: Improve Gray World algorithm parameters: use Gaussian attenuation function, standard deviation σ=1.5, compensation radius r=3.

Decomposition level parameter setting: Retinex multi-scale decomposition level: Level 3

Window size: Gaussian filtering parameters at all levels: Layer 1 (coarse scale): Window size 31 × 31 (capturing global illumination). Layer 2 (Medium Scale): Window size 15 × 15 (Medium Structure). Layer 3 (fine scale): Window size 7 × 7 (retaining high-frequency details).

Running time: Running time (single 1080P image): CPU mode: approximately 0.8 seconds. GPU acceleration: about 0.3 seconds (achieving bilateral filtering and Retinex decomposition). Multi scale decomposition adopts multi threading, and each level of filtering is independently calculated.

7. Please also correct grammatical errors in the paper as there were a few.

Re: The entire text has been checked and grammar errors have been corrected.

---

## [Decision Letter · Decision Letter 1]

19 Aug 2025

PONE-D-25-02402R1Visual Edge Feature Enhancement of Product Appearance Design Images Based on Improved Retinex AlgorithmPLOS ONE

Dear Dr. Chen,

Thank you for submitting your manuscript to PLOS ONE. After careful consideration, we feel that it has merit but does not fully meet PLOS ONE’s publication criteria as it currently stands. Therefore, we invite you to submit a revised version of the manuscript that addresses the points raised during the review process.

We look forward to receiving your revised manuscript.

Kind regards,

Yongjie Li

Academic Editor

PLOS ONE

Journal Requirements:

Additional Editor Comments:

As suggested by one reviewer, please include pseudo-code summarizing the core steps of the algorithm in the Methods section.

Reviewers' comments:

Reviewer's Responses to Questions

**Comments to the Author**

1. If the authors have adequately addressed your comments raised in a previous round of review and you feel that this manuscript is now acceptable for publication, you may indicate that here to bypass the “Comments to the Author” section, enter your conflict of interest statement in the “Confidential to Editor” section, and submit your "Accept" recommendation.

Reviewer #3: (No Response)

Reviewer #4: All comments have been addressed

2. Is the manuscript technically sound, and do the data support the conclusions?

Reviewer #3: Yes

Reviewer #4: (No Response)

3. Has the statistical analysis been performed appropriately and rigorously? 

Reviewer #3: Yes

Reviewer #4: I Don't Know

4. Have the authors made all data underlying the findings in their manuscript fully available?

Reviewer #3: Yes

Reviewer #4: No

5. Is the manuscript presented in an intelligible fashion and written in standard English?

Reviewer #3: Yes

Reviewer #4: Yes

6. Review Comments to the Author

Reviewer #3: I would like to thank the authors for submitting this outstanding paper, which presents an innovative method for enhancing image clarity and improving edge features in product appearance design using an improved Retinex algorithm. The proposed method appears effective and yields impressive results based on the metrics evaluated, such as PCQI, IQE, and IQM.

Regarding research ethics, I did not notice any concerns related to dual publication or violations of research ethics in this work. Upon reviewing the paper, it seems that all data collection and analysis were conducted transparently and in adherence to academic standards. There is no indication of this work being published elsewhere, and I believe the research makes a valuable contribution to the field of image enhancement and analysis.

I recommend the acceptance of this paper, with the continued commitment to transparency in future related publications.

Reviewer #4: I appreciate the substantial improvements you have made in revising the manuscript. For reproducibility, please include pseudo-code summarizing the core steps of your improved algorithm in the Methods section.

7. PLOS authors have the option to publish the peer review history of their article (what does this mean? ). If published, this will include your full peer review and any attached files.

**Do you want your identity to be public for this peer review?** For information about this choice, including consent withdrawal, please see our Privacy Policy .

Reviewer #3: No

Reviewer #4: No

---

## [Author Response · Author response to Decision Letter 2]

25 Aug 2025

Amendment notes

Journal Requirements:

Additional Editor Comments:

As suggested by one reviewer, please include pseudo-code summarizing the core steps of the algorithm in the Methods section.

Re: The core algorithm steps of the visual edge feature enhancement method for the appearance design image of the above product are pseudocode, as shown in Figure 2.

Reviewers' comments:

1. If the authors have adequately addressed your comments raised in a previous round of review and you feel that this manuscript is now acceptable for publication, you may indicate that here to bypass the “Comments to the Author” section, enter your conflict of interest statement in the “Confidential to Editor” section, and submit your "Accept" recommendation.

Reviewer #3: (No Response)

Reviewer #4: All comments have been addressed

2. Is the manuscript technically sound, and do the data support the conclusions?

Reviewer #3: Yes

Reviewer #4: (No Response)

3. Has the statistical analysis been performed appropriately and rigorously?

Reviewer #3: Yes

Reviewer #4: I Don't Know

4. Have the authors made all data underlying the findings in their manuscript fully available?

Reviewer #3: Yes

Reviewer #4: No

Re:The statistical data in Table 1 are the mean values of PCQI, IQE, and IQM data for different methods. We have provided additional measurement data points for median and variance based on your feedback, as shown in Table 1.

5. Is the manuscript presented in an intelligible fashion and written in standard English?

Reviewer #3: Yes

Reviewer #4: Yes

Re: Thank you for your evaluation. I have thoroughly reviewed the paper and provided the corresponding content in response to your project feedback mentioned above.

6. Review Comments to the Author

Reviewer #3: I would like to thank the authors for submitting this outstanding paper, which presents an innovative method for enhancing image clarity and improving edge features in product appearance design using an improved Retinex algorithm. The proposed method appears effective and yields impressive results based on the metrics evaluated, such as PCQI, IQE, and IQM.

Regarding research ethics, I did not notice any concerns related to dual publication or violations of research ethics in this work. Upon reviewing the paper, it seems that all data collection and analysis were conducted transparently and in adherence to academic standards. There is no indication of this work being published elsewhere, and I believe the research makes a valuable contribution to the field of image enhancement and analysis.

I recommend the acceptance of this paper, with the continued commitment to transparency in future related publications.

Re:Thank you for your review and affirmation.

Reviewer #4: I appreciate the substantial improvements you have made in revising the manuscript. For reproducibility, please include pseudo-code summarizing the core steps of your improved algorithm in the Methods section.

Re: The core algorithm steps of the visual edge feature enhancement method for the appearance design image of the above product are pseudocode, as shown in Figure 2.

7. PLOS authors have the option to publish the peer review history of their article (what does this mean?). If published, this will include your full peer review and any attached files.

Do you want your identity to be public for this peer review? For information about this choice, including consent withdrawal, please see our Privacy Policy.

Reviewer #3: No

Reviewer #4: No

---

## [Editor Report · Decision Letter 2]

28 Aug 2025

Visual Edge Feature Enhancement of Product Appearance Design Images Based on Improved Retinex Algorithm

PONE-D-25-02402R2

Dear Dr. Chen,

We're pleased to inform you that your manuscript has been judged scientifically suitable for publication and will be formally accepted for publication once it meets all outstanding technical requirements.

Within one week, you'll receive an e-mail detailing the required amendments. When these have been addressed, you'll receive a formal acceptance letter and your manuscript will be scheduled for publication.

Kind regards,

Yongjie Li

Academic Editor

PLOS ONE
---

## [Editor Report · Acceptance letter]

PONE-D-25-02402R2

PLOS ONE

Dear Dr. Chen,

I'm pleased to inform you that your manuscript has been deemed suitable for publication in PLOS ONE. Congratulations! Your manuscript is now being handed over to our production team.

Kind regards,

on behalf of

Professor Yongjie Li

Academic Editor

PLOS ONE